# Risk Assessment of Miners’ Unsafe Behaviors: A Case Study of Gas Explosion Accidents in Coal Mine, China

**DOI:** 10.3390/ijerph16101765

**Published:** 2019-05-18

**Authors:** Ruipeng Tong, Yunyun Yang, Xiaofei Ma, Yanwei Zhang, Shian Li, Hongqing Yang

**Affiliations:** 1School of Emergency Management and Safety Engineering, China University of Mining and Technology (Beijing), Beijing 100083, China; yangyunyun0225@126.com (Y.Y.); 18811739823@163.com (X.M.); 13121957417@163.com (Y.Z.); 2Safety Center, Research Institute of Highway Ministry of Transport, Beijing 100088, China; sa.li@rioh.cn (S.L.); hq.yang@rioh.cn (H.Y.)

**Keywords:** gas explosion, unsafe behaviors, probabilistic risk assessment, Monte Carlo simulation, sensitivity analysis, behavior-based safety (BBS)

## Abstract

China’s coal mine production situation is grim and various types of accidents occur frequently, and gas explosion accidents are the highest incidence of coal mine accidents. The authors selected 200 gas explosion accidents of coal mine enterprises in recent years, and extracted a large number of workers’ unsafe behaviors. Meanwhile, four working types related to gas explosion accidents were obtained, namely ventilation, gas prevention and fire extinguishing, blasting, and electrician. This article listed some influencing factors of unsafe behaviors and corrected the probability of unsafe behaviors. In addition, a probabilistic risk assessment model was established, and the Monte Carlo method was used to analyze the risks caused by unsafe behaviors of various working types. The results show that the risk of unsafe behaviors caused by the ventilation working type is the highest, followed by gas prevention and fire extinguishing, and finally blasting and electrician. This paper studies the influencing factors of miners’ unsafe behaviors from the perspective of behavior, guarantees effectively the safety management of coal mine enterprises, and lays a foundation for studying unsafe behaviors related to coal mine gas explosions.

## 1. Introduction

Coal mines have always played a vital role in promoting the development of various countries. they are an important energy source for countries in the world. In China, coal is the pillar industry of economic development, and it represents economic development prospects and provides important guarantees for China’s economic and social development and energy supply security [1,2]. China’s coal production capacity accounts for 30% of global production capacity and is also the largest coal-consuming country, with coal consumption accounting for about 60% of China’s energy consumption. At the same time, the coal industry has remained in an irreplaceable position for a long time, and China’s coal-based energy situation will not change fundamentally [3,4]. Nevertheless, the coal mine accident casualty rate in China has reached more than 70% of the global total. [5]. Complex working conditions, the harsh working environment, and many dangerous factors have formed a severe production situation in coal mines [6,7]. This situation has not only caused adverse social impacts, but also widespread concern in the international community, directly affecting the image of the Chinese government and foreign trade of enterprises. Among the many types of accidents, gas explosions are considered to be the most serious type of accident in coal mines [8,9,10]. Not only for Chinese coal mines, but also for many other countries, gas explosions are a serious threat [11,12,13,14,15]. For example, mining is a very important industry in Poland. One of the most common hazards in a coal mine accident is methane, so the fatality rate of gas explosion accidents is very high. When the situation is serious, it may account for half of the number of coal mine accidents in the same period. Similarly, gas explosion accidents in some countries such as Russia and the Ukraine are also serious. Figure 1 is based on the “Statistical Yearbook of China’s Coal Industry”, which calculates the percentage of deaths caused by several different types of accidents in China’s coal mines from 2001 to 2018. It is obvious that gas explosion accidents are more likely to occur than other accidents [16].

This paper shows statistics data on the occurrence of coal mine accidents in China from 2001 to 2018 and presents them in the form of a histogram. It can also be seen in Figure 2 that gas explosion accidents have been declining year by year. The reasons for this situation are that more attention has been paid to mine accidents, and more advanced equipment and technology have been invested in China. Further, Coal mine accidents around the world have been well controlled and prevented, and many countries have adopted advanced coal mine production equipment and technology to ensure the safety of miners. However, for coal mine production accidents in China, gas explosion accidents are still a prominent problem in the field of coal mine safety and occupy an absolute proportion in major and special mine accidents [2,17]. Therefore, in order to prevent the occurrence of gas explosion accidents, it is necessary to go deep into the accident-causing mechanisms and consider the causes of accidents [18]. The three basic elements of a gas explosion accident are gas, ignition source, and sufficient oxygen. The generation of the fire source is controllable, and the causes are almost all related to human activities [19,20]. Simultaneously, some studies have proved that the proportion of human factors causing gas explosion accidents is about 97% [21]. Therefore, reducing people’s unsafe behaviors has become the key to preventing gas explosions [22]. However, the actual situation of gas explosion accident research is that more than 80% of scientific research projects aim to solve the unsafe state of objects and fewer than 20% of scientific research projects aim to study the behavioral causes of accidents. Some scholars have investigated coal mine gas explosions and determined that the accidents occurred because of unsafe human behaviors. For example, Meng et al. [23] believed that many unsafe behaviors would lead to gas explosions and made a risk assessment of unsafe behaviors. Yin et al. [10] counted the gas explosion accidents from 2000 to 2014 and clearly analyzed the unsafe behaviors. The above studies show that safety must be considered from the perspective of unsafe human behaviors in order to prevent gas explosion accidents [8].

Risk assessment is an important means to study unsafe behaviors [24]. It is the main research content in the field of coal mine safety. Meanwhile, it is also important to reduce gas explosion accidents by conducting a risk assessment of unsafe behaviors [25,26]. However, many studies only consider the assessment of unsafe behaviors of coal mine systems, but do not assess the risk of each working type [27,28]. It can also be said that most of these studies assess risks from a systematic point of view [29,30,31], and few studies have been conducted to assess the risk of unsafe behaviors involved in a type of disaster [32]. In this study, 200 gas explosion accidents were selected, and unsafe behaviors were extracted and classified; then, the key working types related to gas explosion were summarized. The accident losses will be quantified, and a risk assessment model will be established based on the Monte Carlo method to assess the risk of different working types. It is worth noting that the risk assessment method used in this paper is probabilistic risk assessment. It can comprehensively characterize the distributions and values of parameters and directly deal with the uncertainty of each risk factor, which will make the results more scientific and credible. The results of this paper can help us better understand the risks of unsafe behaviors associated with gas explosions and help to provide a basis for effective control of unsafe behaviors in gas explosion accidents.

## 2. Research Foundation

### 2.1. Selection of Key Working Types

Coal mine production involves many different working types of miners; however, there are no specific standards and regulations for the division of working types in the coal industry. It is difficult for coal mining enterprises to accurately define the work content of relevant workers. Simultaneously, the training of miners’ skills and accident prevention education are also lacking [33]. Therefore, it is necessary to clearly define the working types. This method is significant because the operators can clarify the content of work and job responsibilities; some underground coal mine knowledge and other skills training can make the design and implementation of courses more targeted; and different working types can be carried out in safety training to achieve self-rescue in distress. Through understanding the situation of coal mining enterprises in Hebei, Henan, Shanxi, Inner Mongolia, and other provinces, six types of working related to gas accidents in coal mines were summarized according to underground operation modes, as shown in Table 1.

The authors studied the unsafe behaviors of workers related to gas explosions from the perspective of working types and listed the above six working types. However, the four categories of ventilation, gas prevention and fire extinguishing, blasting, and electrician are the key working types of accidents with high frequency of accidents. Therefore, the key working types selected in this paper are ventilation, gas prevention and fire extinguishing, blasting and electrician. The unsafe behaviors of these key working types were taken as the research object to conduct behavioral risk assessment.

### 2.2. Classification of Unsafe Behaviors

#### 2.2.1. Unsafe Behaviors of Key Working Types

(1) Ventilation

Gas accumulation is an important cause of gas explosions, and mine ventilation is an important method to prevent gas accumulation [34,35,36]. At the same time, due to the complex working environment of coal mines in China, there are many toxic and harmful gases in addition to regular gas. Therefore, the ventilation system plays an important role in preventing gas explosion accidents [37]. When performing underground coal mine operations, the ventilation system often creates some problems, such as insufficient supply of mine air, chaotic ventilation management, unreasonable ventilation systems, and a blockage of the roadway. If these problems are not discovered or solved in time, it is easy to cause gas accumulation and gas overruns, which eventually lead to gas explosions. The main functions of the ventilation working types are to monitor the operation and maintenance of the ventilator in time and pay attention to the air volume in the roadway. Attention should be paid to the formulation of mine ventilation measures to avoid the occurrence of unsafe behavior. The types of ventilation work include many jobs, such as a mine wind measuring worker, ventilator operator, air duct installer, etc. However, the causes for the occurrence of gas overrun and gas accumulation in the underground are mostly due to the unsafe working behaviors and unsafe working modes of the workers in these positions. Therefore, it is very important to study the unsafe behaviors in ventilation workers.

(2) Gas Prevention and Fire Extinguishing

The work of “one ventilation, three prevention” has always been given more attention by Chinese coal mining enterprises, including ventilation, gas prevention, fire prevention, and coal dust prevention [38,39]. This paper does not assess the content of coal dust, and mainly studies the first three categories. The contents of the ventilation part have been mentioned before, so the contents of gas prevention and fire extinguishing are summarized together. As far as we know, the main causes of gas accumulation are improper gas emissions, gas emission anomalies and gas concentration overruns, all of which are caused by unsafe human behaviors [40]. The main job of this working type is to regularly detect the gas concentration at each working place in the underground and then formulate measures to control it. In addition, it is necessary to pay attention to the management of the goaf to prevent open flames. Therefore, it is of great significance to study the unsafe behaviors of this working type for guaranteeing the safety of underground production and the safety of miners. The working type of gas prevention and fire extinguishing also includes many jobs, such as gas inspectors, mine firefighters, gas drainage workers, etc.

(3) Blasting

Blasting operations are the most prone to sparks. Gas explosion is very likely to occur when the blasting personnel carry out blasting operations under the condition of gas accumulation and exceed the limits. Therefore, avoiding sparks during the blasting process is important for preventing gas explosions [41]. Many accidents were caused by workers’ incorrect operations. The unsafe behaviors of the blasting workers during the blasting operation provided the breakthrough of this research. The measures to prevent the blasting workers from sparking during the blasting process are to not perform blasting operations in the environment of gas accumulation and excess, and to strictly control blasting behavior during the blasting process to avoid sparks. Types of blasting work include underground blaster, mine gunpowder warehouse workers, and explosive material escort workers. The strict prevention of gas explosions caused by sparks in blasting operations and unsafe human behaviors required consideration.

(4) Electrician

Electricity is absolutely indispensable to coal mining enterprises nowadays. At the same time, the reliability of electromechanical equipment is an important guarantee for the safe production of coal mines. However, the unsafe safety protection devices, unsafe maintenance methods, and incorrect operation of mechanical and electrical equipment will lead to coal mine accidents. The task of electrician working types is to ensure the normal operation of coal mine electrical equipment and to ensure the normal supply of electricity without affecting production. From the perspective of the possibility of gas explosions, the unsafe behaviors of electrician working types mainly include improper operation, which generates electric sparks, and other unsafe behaviors that can lead to gas accumulation, such as the failure of local ventilators due to power outages, which further results in gas accumulation. Therefore, the study of unsafe behaviors caused by this working type can be targeted to formulate technical safety measures that have practical significance for preventing gas explosion accidents. Electrician jobs include mine electromechanical installers, mine electromechanical maintenance electricians, power distribution workers, and welding workers.

The unsafe behaviors or possible unsafe behaviors that often occur in the above four key working types are summarized in Table 2. This table, which is of reference value, is a comprehensive list of these unsafe behaviors.

#### 2.2.2. Classification of Unsafe Behaviors

There are 13 types of unsafe behaviors stipulated in Appendix A of GB 6441-86 [42]. The contents are as follows:(1)Operating incorrectly, ignoring safety and warning (OIW);(2)Failure of safety device (FSD);(3)Use of unsafe devices (UUD);(4)Hand instead of tool operation (HIT);(5)Improper storage of objects (ISO);(6)Venture into dangerous places (VDP);(7)Climbing and sitting in unsafe positions (CSU);(8)Work and stay under lifting objects (WSI);(9)Repair, inspection, welding, cleaning and other operations are carried out while the machine is running (RIW);(10)Workers have distracted behaviors (WHD);(11)In the workplace where personal protective equipment must be used, the use of personal protective equipment is neglected (PPE);(12)Unsafe attire (UA);(13)Mishandling of Flammable and Explosive Dangerous Goods (MFE).

Taking 200 coal mine gas explosion accidents from 2001 to 2015 as samples. And according to the unsafe behaviors categories specified in the national standard, the unsafe behaviors of four key types of ventilation, Gas prevention and fire extinguishing, blasting, and electrician are classified. The results obtained are shown in Table 3.

## 3. Research Methods

### 3.1. Probability of Unsafe Behaviors

The probability of unsafe behavior is critical for risk assessment [43]. However, many studies on unsafe behavioral assessments have been directly classified and weighted in the past. Some studies divide unsafe human behaviors into two parts: organizational behavior and individual behavior. By establishing an evaluation index system and using the measuring scale to weigh the frequency of unsafe behaviors, the occurrence probability of unsafe behavior is finally determined. The results of these studies are subjective, one-sided, and uncertain. At the same time, studies have shown that the frequency of calculation also needs to take into account other factors that affect the occurrence of unsafe behaviors [44], such as personal factors, organizational factors, and environmental factors [43,45]. The relationship between these factors is difficult to quantify, and the fuzzy decision-making method has a good effect on quantifying many factors and reducing human subjectivity. This paper does not elaborate on the fuzzy decision-making method.

Based on the research of Dan et al. [44], the influencing factors of coal mine workers’ unsafe behaviors were divided into organizational factors, human-machine factors, and personal factors in this paper. As shown in Figure 3, the organizational factors are divided into a rules and regulations factor, an education and training factor, and a safety reward factor; human-machine factors are divided into a workload factor, a device factor, and a working environment factor; personal factors are divided into a safety physiological and psychological factor, and a skills and knowledge factor.

Similarly, based on the research foundation of Dan et al. [44], the model of the relationship between the influencing factors was obtained, and the occurrence probability of unsafe behavior was revised. The formula is as follows:(1)$$F\left(x\right)=-0.0167\left[\left(0.2a+0.6b+0.2c\right)+\frac{d+e+f}{3}+\frac{g+h}{2}\right]+1.5.$$
(2)$$P=P^{\prime }\times F\left(x\right)$$
In the formula, F(x) is the revision factor, a is the rules and regulations factor, b is the education and training factor, c is the safety reward factor, d is the workload factor, e is the device factor, f is the working environment factor, g is the safety physiological and psychological factor, and h is the skills and knowledge factor. P is the revised probability of unsafe behavior, and P’ is the probability of unsafe behavior.

These parameters are dimensionless and subject to triangular distribution. However, for the types of work related to coal mine gas explosion, the value range of different working types is discrepant. According to the actual situation of Chinese coal miners and some reference materials [19,23], the values’ ranges of these influencing factors are listed in Table 4 and Table 5.

Since the different positions of coal miners have different characteristics, the values’ ranges of these factors are discrepant. For example, for the blasting working type, rules and regulations, education, and training are detailed and sufficient, so the range of a and b is [1], but the safety reward is not perfect; therefore, the value is [1–3], and the workload is relatively moderate, so the value is also [1–3]. These values were determined according to the specific working conditions of coal miners, and the correction factors obtained were also reliable.

### 3.2. Quantification of Accident Hazards

Quantification of the consequences of coal mine gas explosion accidents is equally important for risk assessment [46]. The occurrences of gas explosions in coal mines are not accidental, but the result of the combined actions over time and space via dangerous factors in underground mines. Generally, coal mine enterprises start from aspects of safety management and technical improvement in the prevention of accidents [47], but the influence of human factors is the most important in the occurrence of accidents. Therefore, it is necessary to study the possibility of accidents caused by unsafe human behaviors and quantify the degree of accidents caused by unsafe behaviors [48,49]. As far as we know, unsafe human behaviors may or may not lead to accidents. Studying the probability of accidents caused by unsafe behaviors is a critical step to quantify the consequences of accidents. In the production process of coal mine operations, human behaviors can be quantified through reliability [50], while also taking into account the reliability of equipment and the relationship between people and equipment. These indicators are all possible causes of gas explosion accidents.

Considering the actual situation of coal mines, Table 6 lists the probability scores of accidents caused by unsafe behavior, which can be divided into four grades.

According to the “Coal Mine Production Safety Accident Report and Investigation and Treatment Regulations”, and the actual situation of gas explosion, gas explosion accidents are divided into the following five grades [8]. The severity of the accident is specified, and the specific accident loss and grade, as well as the scoring criteria, are listed in Table 7. For the consequences of gas explosions caused by unsafe behaviors, this paper uses the accident hazard index for measurement—that is, the product of the possibility that unsafe behavior may cause accidents and the severity of accident losses.

### 3.3. Behavioral Risk Assessment Model

Many scholars have studied the risks of coal mine accidents, but there are few risks in studying unsafe behaviors. The risk value obtained by the traditional risk method is the product of the probability of accident occurrence and the consequences of accidents [51,52]. Based on the above contents, this paper conducts a risk assessment of the unsafe behaviors related to coal mine gas explosions. The evaluation model obtained is as follows:(3)R=P×I
(4)I=B×C
where R is the risk of unsafe behavior, P is the revised probability of unsafe behavior, I is the accident hazard index, C is the severity of accident losses, B is the probability of an accident with a serious consequence C caused by unsafe behavior.

## 4. Result and Discussion

### 4.1. Unsafe Behavior Risk Assessment Results

#### 4.1.1. The Possibility and Hazard Index of Unsafe Behavior

In this study, Crystal Ball software 11.1 (ORACLE, Redwood City, CA, USA) was used to test the goodness-of-fit test for data, and the distributions of these factor’s parameters were more consistent. The results show that the parameter’s distributions of the four key working types related to coal mine gas explosion accidents are most consistent with triangular distribution, and the probabilities of unsafe behaviors are shown in Table 8.

From the results in Table 8, it can be seen that ventilation is the most common type of work to cause gas explosion accidents, both from the point of view of the occurrence of unsafe behavior and from the perspective of the revised occurrence probability. The possible reason for the results in this table is that ventilation is the most important factor for preventing gas explosions in coal mines. Ventilation problems can occur due to faulty ventilating equipment and the unsafe behavior of workers. The number of unsafe behaviors in gas prevention and fire extinguishing ranks second, because the basic element of a gas explosion accident is the ignition source, and the workers in the gas prevention and fire extinguishing working types have imperfect preventive measures, which will be likely to lead to a gas explosion accident. The probabilities of unsafe behavior in the blasting and electrician working types are relatively small. The reason for this difference in results may be that the blasting equipment and electrical equipment in modern coal mines are relatively advanced, which can effectively guarantee the safety of coal mine production and miners.

According to the relevant data of coal mine gas explosions and the actual production situation of coal mines, the authors analyzed the possibility and severity of gas explosion accidents caused by unsafe behaviors and determined the value of the accident hazard index. At the same time, the values were tested again by the Crystal Ball software 11.1; meanwhile, the accident hazard index obeyed normal distribution. The results are shown in Table 9.

#### 4.1.2. Risk Value of Unsafe Behavior

According to Formulas (3) and (4) for the calculation of the risk value of unsafe behaviors, the Monte Carlo method is used to carry out a statistical simulation on the risk model composed of the unsafe behaviors’ revised probability for the key working types and the value of the accident hazard index, setting the maximum number of simulations to 10,000, with a confidence interval of 95%. Simultaneously, on the basis of the probabilistic risk assessment model of unsafe behaviors, the risk of unsafe behaviors in each post is obtained. The results are shown in Table 10, and the risk values show lognormal distributions.

It can be seen from the table that for the key working types related to gas explosion accidents, the unsafe behavior risk value of the ventilation type is the largest. Among these risks, the risks of OIW and UUD unsafe behaviors are relatively high at 4.35 × 10^−1^ and 1.51 × 10^−1^ respectively. Secondly, the risk of unsafe behaviors exists in the blasting working type, and the risk of unsafe behaviors in types UUD and OIW are higher at 2.25 × 10^−1^ and 1.49 × 10^−1^. In third place is the risk value of gas prevention and fire extinguishing, and the risk of OIW unsafe behaviors is higher. Finally, the risk value of the electrician working type is still the highest risk of unsafe behavior in OIW.

From the perspective of the type of unsafe behaviors, the occurrence frequency and possibility of the OIW and UUD types of unsafe behaviors are higher. For example, the risk values of ventilation, gas prevention and fire extinguishing, blasting, and electrician for OIW unsafe behavior are 4.35 × 10^−1^, 3.35 × 10^−1^, 1.49 × 10^−1^ and 9.96 × 10^−2^, respectively. The risk value caused by the unsafe behavior of UUD is also relatively high, and the risk value caused by the other unsafe behaviors is relatively low; these values are not listed one by one. However, it is worth noting that the impacts of different types of unsafe behaviors among these four key working types are distinguishing. That is to say, if measures are taken to avoid the occurrence of gas explosion accidents, it is necessary to take precautions against the characteristics of different working types.

Figure 4 shows the unsafe behaviors’ risk in four key working types; this figure contains the minimum risk, quarter, median, three-quarters and maximum risk value for each type of work, and characterizes the distribution of risk. The results of Figure 3 can also support the conclusion above. The risk values of the four working types occur in the order of ventilation > gas prevention and fire extinguishing > blasting > electrician. Ventilation types have the greatest risk, while the risk values for each type of unsafe behavior risk for the electrician type show little difference, thus producing the results shown in the figure.

Based on the above results, the ventilation in coal mines is the key to preventing gas explosion accidents. For the unsafe ventilation behaviors, it is recommended to check the ventilation facilities in time and operate the ventilation machines according to the regulations. Avoiding unsafe behaviors during actual operations is critical to preventing gas accumulation and reducing gas anomalies in underground mines. Undoubtedly, it is also important to prevent gas prevention and fire extinguishing, as well as blasting and electrical operations, whose purpose is to control the generation of fire sources. Mine workers should strengthen the monitoring and inspection of gas and fire sources to avoid gas explosions.

### 4.2. Sensitivity Analysis

Sensitivity analysis can obtain the parameters that affect the results of risk assessment, and then one can take appropriate measures to improve the factors related to each parameter [53,54]. Figure 5 shows the sensitivity of each factor.

The accident loss index has the greatest impact on the risk results, reaching 43.71%, while the sensitivities of the remaining parameters are relatively small, including parameters a, b, c, d, e, f, g, h, all of which are negatively sensitive and have a negative impact on the results. Among these above parameters, the effects of f (working environment factor) and d (workload factor) are large at −14.13% and −12.11%, respectively. Secondly, h (skill knowledge factor) and c (safety reward factor) also had moderately negative effects on the results, which were −9.21% and −8.24% respectively. Finally, a (regulatory factors), b (educational training factors), g (safety physiology and psychological factors), and e (equipment factors) had less negative impact on risk outcomes. The results are also consistent with those of the studies on the probability of unsafe behaviors in key working types.

Through the analysis of the above data, several points can be obtained: the same parameter has distinct effects on workers in different working types. Therefore, to avoid the occurrence of coal mine gas explosion accidents, it is necessary to design diverse methods and measures in a targeted manner; because of the great influence of the working environment and workload, coal mine enterprises should improve the working environment and change the workload according to the working characteristics of different working types. For example, because blasting workers are in a dangerous working environment with noise and pollution for a long time, they need to improve their protective equipment, or switch their jobs in a timely manner, so as to avoid causing coal miners to be engaged in the same kind of work all the time.

### 4.3. Uncertainty Analysis

The results of this study have theoretical and practical significance. Previous studies have rarely conducted risk assessment of unsafe behavior, but this paper selected 200 gas explosion accidents from the perspective of unsafe behaviors risk and analyzed the unsafe behaviors involved. This study is representative and provides ideas and methods for the risk assessment of unsafe behaviors of other accidents in coal mines.

In the probabilistic risk assessment process, the sample size is very important and will have a great impact on the assessment results, but the research in this paper may have an impact on the results because of the insufficient sample size. From the samples collected in this paper, the author summarizes four key working types based on the unsafe behaviors extracted. If more samples are collected, the number of unsafe behaviors and the key working types will be greater. Secondly, the selection of influencing factors of unsafe behaviors and the relationship between factors will lead to uncertainty. Although the relationships between various factors have been obtained on the basis of previous research, these relationships have not been verified, which is also an important reason for uncertainty. Finally, the probability of accidents caused by unsafe behaviors and the quantitative index of accident losses are also obtained and calculated by experts’ scores and questionnaires. All of the above may make the results subjective.

Further research and analysis is still needed to address these uncertainties. For example, the big data method is used to identify and collect unsafe behaviors in order to reduce the uncertainty caused by the sample size of unsafe behaviors [55]. Simultaneously, the factors and relationships that affect the unsafe behavior of coal miners are deeply considered, and the influencing factors are deeply normalized and weighted. In addition, it is necessary to further determine the accident loss quantification index.

### 4.4. Unsafe Behavior Risk Management

This article takes coal mine gas explosion accidents as the research background and evaluates the risk of the unsafe behaviors of coal miners. The risk results are provided by the probability of workers’ unsafe behaviors and the loss of accidents caused by unsafe behaviors. The preceding part of the article has shown that the occurrence of unsafe behaviors is related to personal factors, organizational factors, and environmental factors, and the number of unsafe behaviors in these four key working types is in the order of ventilation > gas prevention and fire extinguishing > blasting > electrician. Coal mine enterprises should develop corresponding measures to prevent the occurrence of gas explosion accidents according to the characteristics of different working types. As far as we know, for coal mining companies, safety culture and safety climate are indispensable supports for the continuous realization of safety production. Because only relying on scientific and technological means cannot achieve real safety, safety culture and scientific management methods are needed to make workers clearly understand their responsibilities and prevent accidents. Scientific and effective safety management methods, specific safety management objectives, favorable safety production environments, safety culture, and safety climate form a safety management system. A perfect safety management system is constructed by management and workers, whose function is to improve the production performance of enterprises and reduce the occurrence of unsafe behaviors. However, the most important measure is to curb the occurrence of unsafe behaviors.

The most effective study of unsafe behaviors is Behavior-Based Safety (BBS), which is based on human factor engineering and behavior-based theory and gradually builds into a systematic safety management system [56]. BBS mainly conducts research on unsafe human behaviors and aims at cultivating people’s safety awareness as the main perspective to continuously improve the overall safety level [57]. By exploring the rules of behaviors development, qualitative analysis and quantitative measurement of behaviors can be carried out, and then the method for correcting behaviors can be standardized. Implementing BBS management processes can improve miners’ thinking and behavior modes to achieve the effect of accident prevention and improve the safety production performance of coal mine enterprises [21]. BBS not only pays attention to the correction of miners’ unsafe behaviors, but also pays attention to the establishment of safety psychology, which has a positive impact on the working attitude of miners. Ultimately, making behavioral safety management the first priority of the enterprise can essentially reduce the generation of unsafe behaviors.

## 5. Conclusions

This paper assesses risks from the perspective of unsafe human behaviors, which is innovative compared with previous studies. Firstly, 200 coal mine gas explosion accidents were selected to extract unsafe behaviors and key working types. Secondly, the model for calculating the probability of unsafe behaviors was improved, and the probabilities of unsafe behaviors were corrected effectively. At the same time, the quantitative accident loss indexes caused by unsafe behaviors were obtained. Finally, the risk assessment model of unsafe behaviors was established and combined with the Monte Carlo simulation method to obtain the evaluation results. This method filled in the gap of the relevant research fields. The research results can help coal mine enterprises effectively avoid the occurrence of gas explosion accidents and also provide a method for risk assessment that has practical significance for other accidents in coal mines.

The results showed that among the four key working types related to gas explosions, the order of the number of unsafe behaviors occurring is as follows: ventilation > gas prevention and fire extinguishing > blasting > electrician. The risk assessment results were also in the above order. Considering the types of unsafe behaviors, the occurrence probability of unsafe behaviors in OIW and UUD were relatively high in four key working types. Sensitivity analysis showed that the impact of the accident loss index was the biggest, accounting for 43.71%, followed by f (work environment factor) and d (workload factor), which had negative impacts on the results, respectively −14.13% and −12.11%.

According to the results obtained in this paper, coal mine enterprises should attempt to prevent unsafe behaviors according to the characteristics of different working types. For example, in view of the unsafe behaviors of ventilation, it is suggested that ventilation facilities should be checked in time and ventilators should be operated according to regulations. More importantly, scientific and effective safety management methods, specific safety management objectives, favorable safety production environment, safety culture, and safety climate are all important factors in the formation of a sound safety management system. Constructing a complete safety management system is an important measure to effectively improve the performance of coal enterprises and reduce the occurrence of miners’ unsafe behaviors. In a word, this research provides reference value for the safety management of coal mine enterprises.

## Figures and Tables

**Figure 1 ijerph-16-01765-f001:**
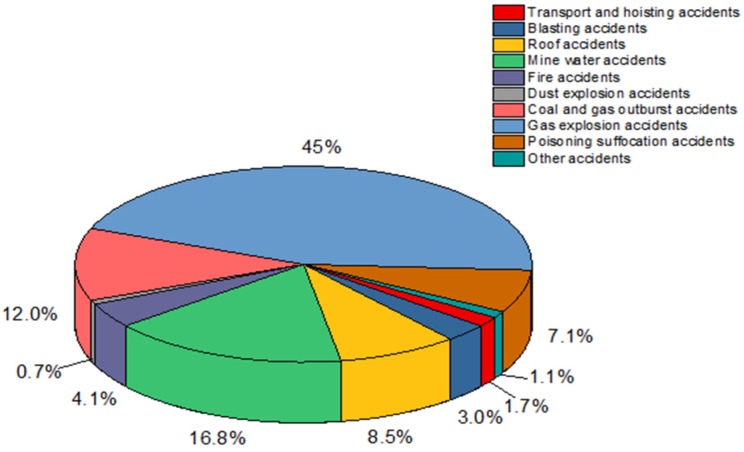
Occurrence of different types of accidents.

**Figure 2 ijerph-16-01765-f002:**
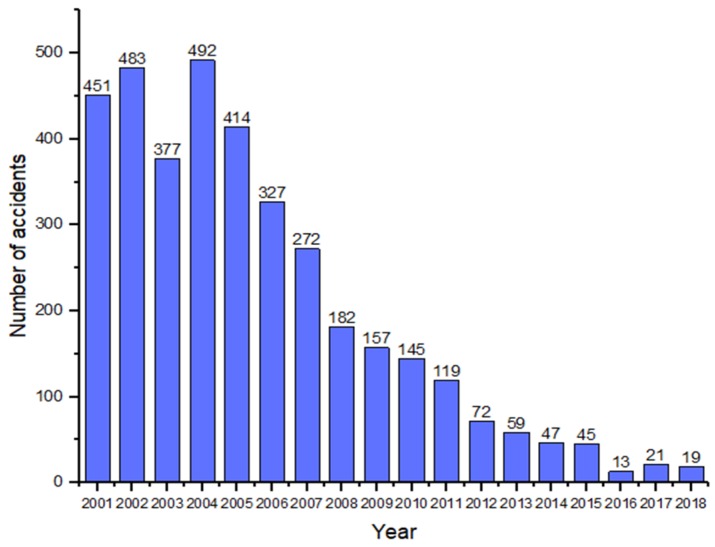
Statistics of gas explosion accidents from 2001 to 2018.

**Figure 3 ijerph-16-01765-f003:**
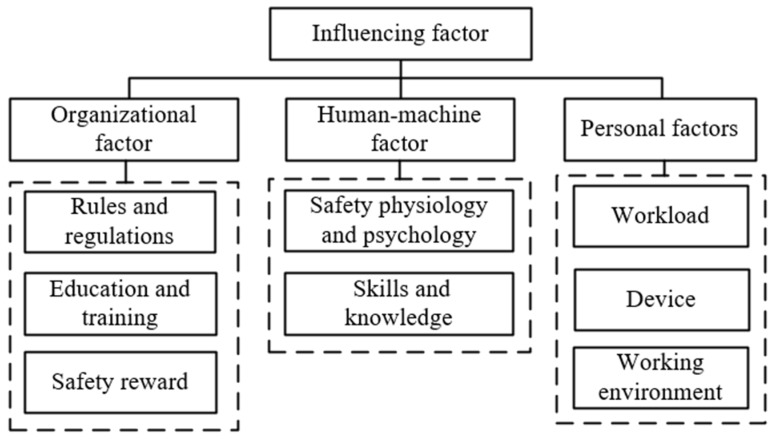
Classification of influencing factors.

**Figure 4 ijerph-16-01765-f004:**
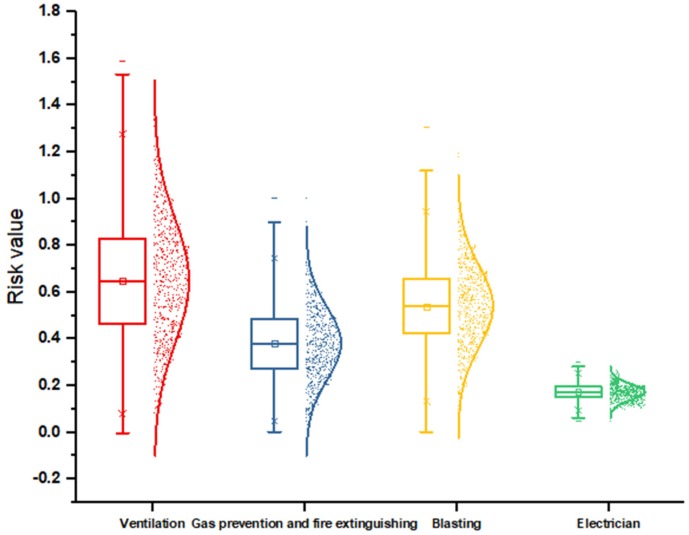
Risks of unsafe behaviors in four key working types.

**Figure 5 ijerph-16-01765-f005:**
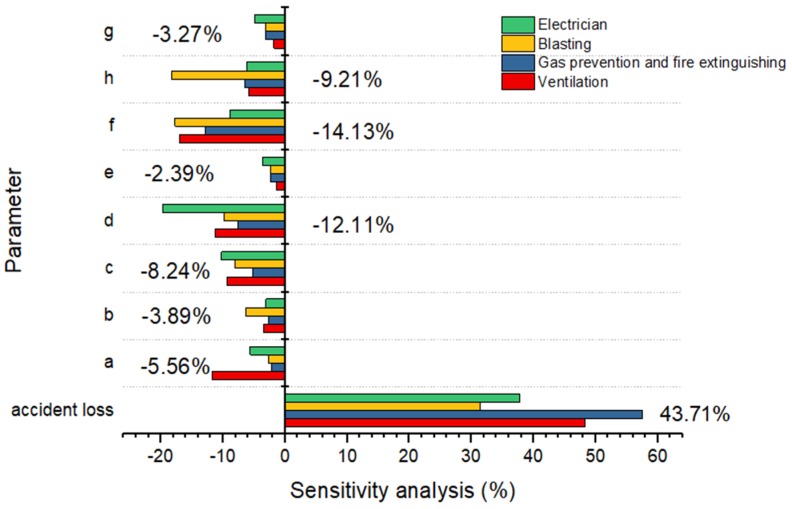
Sensitivity analysis results of influencing factors.

**Table 1 ijerph-16-01765-t001:** Statistics of working types associated with gas explosion accidents.

Serial Number	Working Types	Work Content
1	ventilation	Provide enough fresh air for underground coal mines to meet the oxygen needs of personnel and dilute the gas concentration.
2	gas prevention and fire extinguishing	Detect mine gas and fire and report the situation in time.
3	blasting	Work in accordance with the regulations to avoid sparks in the blasting process.
4	electrician	Install, debug, patrol and repair the mechanical and electrical equipment in coal mines to ensure the normal operation of the mechanical and electrical equipment in the using process.
5	mining	Operate shearers and roadheaders in coal mining and heading faces and be responsible for machine inspection and operation records.
6	transport	Operate hoisting equipment or locomotives in coal mines to transport personnel and materials.

**Table 2 ijerph-16-01765-t002:** Influencing factors of unsafe behaviors in key working types.

Working Types	Unsafe Behaviors
Ventilation	Unqualified ventilation facilities.
	No local ventilator is installed in the heading face.
	Insufficient air supply to the ventilator.
	Local ventilator suction volume is greater than tunnel air volume.
	The local ventilator is installed in the wrong position.
	Using local ventilator instead of main ventilator for ventilation.
	Arbitrary setting of airtight locations leads to windless operation.
	Failure to connect the air duct in time.
	Operating Random air ducts, such as disconnection, connection, hanging inequality.
	Illegal command, neglect and allowing ventilators to turn on and stop local ventilated workers at will in daily operation.
	Turning off the main ventilator at will.
	Conducting forcible production without wind or breeze.
Gas prevention and fire extinguishing	Gas inspectors are not on duty at the scene.
	Not checking gas concentration or insufficient number of times before operation.
	Checked out gas exceeding the limit, but no measures have been taken.
	Did not carry a gas detector or carry an unqualified gas tester when entering the coal mine.
	Gas concentration was not detected under special operation conditions such as drainage and fire operation, etc.
	There is no specific measure for discharging gas.
	Failure to inspect and dispose of blind and abandoned tunnel in time.
	No gas concentration was detected in the airtight area.
	No fire prevention measures were implemented in time for the goaf.
	Non-regular inspection of appropriative gas drainage lanes as required.
Blasting	Gas concentration at blasting site not detected before blasting operation.
	Blasting without wind.
	No ventilation measures were taken after blasting.
	Blasting without special detonator.
	Did not check the clear joints of the blasting busbar.
	Sealing holes are not fully filled with mud.
	Blasting is still under way when the gas concentration is known to exceed the limit.
	Did not check whether the blasthole guarantees the minimum resistance line when blasting.
	It is not confirmed whether the coal body has eliminated outstanding dangers before blasting.
	Not a professional blaster but doing blasting work.
	Blaster doing blasting work at will.
	Short-time continuous blasting in the same operating area
Electrician	Maintenance of electrical equipment under live condition.
	Gas concentration not inspected before overhauling electrical equipment.
	Power supply to electrical equipment in windless areas.
	Power supply to electrical equipment in unknown gas concentration area.
	Unauthorized power outage.
	Use of electrical equipment with cable open joints.
	Use of non-mineral electrical equipment.
	Privately pulling live cables.

**Table 3 ijerph-16-01765-t003:** Classification of unsafe behaviors of key working types related to gas explosion.

Working Types	Classification of Unsafe Behaviors	Characterization	Number of Occurrences
Ventilation	OIW; FSD; UUD; VDP; RIW	Illegal use of equipment, random operation of equipment, neglect of safety and warning, risky operation, still operating when the machine is running, etc.	82
Gas prevention and fire extinguishing	OIW; FSD; UUD; VDP; WHD; MFE	Gas inspectors operate incorrectly, equipment fails, venture into the workplace, gas inspectors do not pay attention to the instrument, do not handle gas, etc.	59
Blasting	OIW; FSD; UUD; VDP; RIW; MFE	Arbitrary blasting, neglect of safety before operation, blasting with non-professional equipment, leaving when blasting operation is in progress, improper handling of explosives, etc.	37
Electrician	OIW; FSD; UUD; VDP; RIW; PPE	Neglecting safety warnings, using unsafe equipment, not wearing personal protective equipment, etc.	22

OIW: Operating incorrectly, ignoring safety and warning; FSD: Failure of safety device; UUD: Use of unsafe devices; VDP: Venture into dangerous places; RIW: Repair, inspection, welding, cleaning and other operations are carried out while the machine is running; WHD: Workers have distracted behaviors; MFE: Mishandling of Flammable and Explosive Dangerous Goods; PPE: In the workplace where personal protective equipment must be used, the use of personal protective equipment is neglected.

**Table 4 ijerph-16-01765-t004:** Range of influencing factors.

	Factor	a	b	c	d	e	f	g	h
Values									
0–1		detailed	enough	many	less	good	good	good	adequate
1–3		general	general	moderate	medium	general	moderate	medium	medium
3–10		deficient	insufficient	few	many	bad	bad	poor	deficient

**Table 5 ijerph-16-01765-t005:** Values of factors for four key working types.

	Parameter	a	b	c	d	e	f	g	h
Working Types									
Ventilation		[0,1]	[0,1]	[3,10]	[3,10]	[0,1]	[3,10]	[1,3]	[0,1]
Gas prevention and fire extinguishing		[0,1]	[0,1]	[1,3]	[3,10]	[0,1]	[3,10]	[1,3]	[0,1]
Blasting		[0,1]	[0,1]	[1,3]	[1,3]	[0,1]	[3,10]	[3,10]	[0,1]
Electrician		[0,1]	[0,1]	[3,10]	[3,10]	[0,1]	[3,10]	[1,3]	[0,1]

**Table 6 ijerph-16-01765-t006:** The possibility of accidents caused by unsafe behaviors.

The Possibility of an Accident	Score
Very likely	1
Probably	0.5
Occasionally	0.1
Unlikely	0.05
Scarcely possible	0.01

**Table 7 ijerph-16-01765-t007:** The severity of the accident.

Degree of Injury	Economic Losses Caused by Accidents (CNY)	Value
One person was slightly injured	Two thousand below	1
More than one person was slightly injured	Two thousand to ten thousand	2
More than one person was seriously injured	Ten thousand to one million	3
One person died	One million to five million	4
More than one person died	More than five million	5

**Table 8 ijerph-16-01765-t008:** Probabilities of unsafe behaviors for key working types.

Key Working Types	Unsafe Behaviors	Number of Occurrences	Occurrence Probability	Correction Factor F(x)	Revised Probability	Percentage of Unsafe Behaviors
Ventilation	OIW	32	0.16	1.376	0.22	41%
	FSD	19	0.095		0.13	
	UUD	15	0.075		0.10	
	VDP	6	0.03		0.04	
	RIW	10	0.05		0.07	
Gas prevention and fire extinguishing	OIW	21	0.105	1.391	0.15	29.5%
	FSD	8	0.04		0.06	
	UUD	17	0.085		0.12	
	VDP	3	0.015		0.02	
	WHD	2	0.01		0.01	
	MFE	8	0.04		0.06	
Blasting	OIW	9	0.045	1.78	0.06	18.5%
	FSD	4	0.02		0.03	
	UUD	13	0.065		0.09	
	VDP	2	0.01		0.01	
	RIW	2	0.01		0.01	
	MFE	7	0.035		0.05	
Electrician	OIW	7	0.035	1.376	0.05	11%
	FSD	2	0.01		0.01	
	UUD	3	0.015		0.02	
	VDP	5	0.025		0.03	
	RIW	2	0.01		0.01	
	PPE	3	0.015		0.02	

**Table 9 ijerph-16-01765-t009:** Values and distributions of the accident hazard index.

Types of Unsafe Behaviors	Ventilation	Gas Prevention and Fire Extinguishing	Blasting	Electrician
OIW	N, 2 ± 1.2	N, 2.25 ± 1.05	N, 2.5 ± 1.6	N, 2 ± 0.45
FSD	N, 0.125 ± 0.045	N, 0.1 ± 0.085	N, 0.5 ± 0.145	N, 0.3 ± 0.125
UUD	N, 1.5 ± 0.6	N, 0.2 ± 0.0.135	N, 2.5 ± 0.75	N, 2.5 ± 1.25
VDP	N, 0.45 ± 0.15	N, 0.1 ± 0.07	N, 2 ± 1.4	N, 0.3 ± 0.15
RIW	N, 0.25 ± 0.19	-	N, 0.25 ± 0.14	N, 0.225 ± 0.11
WHD	-	N, 0.125 ± 0.09	-	-
PPE	-	-	-	N, 0.4 ± 0.16
MFE	-	N, 0.125 ± 0.1	N, 2.5 ± 1.8	-

N: Normal distribution (mean value ± standard deviation).

**Table 10 ijerph-16-01765-t010:** Risk Values of Unsafe Behaviors.

Unsafe Behaviors	Ventilation	Gas Prevention and Fire Extinguishing	Blasting	Electrician
OIW	LN, 4.35 × 10^−1^ ± 2.64 × 10^−1^	LN, 3.35 × 10^−1^ ± 1.57 × 10^−1^	LN, 1.49 × 10^−1^ ± 9.48 × 10^−2^	LN, 9.96 × 10^−2^ ± 2.24 × 10^−2^
FSD	LN, 1.626 × 10^−2^ ± 5.81 × 10^−3^	LN, 5.97 × 10^−3^ ± 5.09 × 10^−3^	LN, 1.50 × 10^−2^ ± 4.33 × 10^−3^	LN, 3.00 × 10^−3^ ± 1.24 × 10^−3^
UUD	LN, 1.51 × 10^−1^ ± 6.00 × 10^−2^	LN, 2.40 × 10^−2^ ± 1.62 × 10^−3^	LN, 2.25 × 10^−1^ ± 1.13 × 10^−1^	LN, 5.04 × 10^−2^ ± 2.49 × 10^−2^
VDP	LN, 1.81 × 10^−2^ ± 5.89 × 10^−3^	LN, 1.99 × 10^−3^ ± 1.40 × 10^−3^	LN, 1.98 × 10^−2^ ± 1.41 × 10^−2^	LN, 9.08 × 10^−3^ ± 4.51 × 10^−3^
RIW	LN, 1.74 × 10^−2^ ± 1.33 × 10^−2^	-	LN, 2.49 × 10^−3^ ± 1.40 × 10^−3^	LN, 2.25 × 10^−3^ ± 1.10 × 10^−3^
WHD	-	LN, 1.23 × 10^−3^ ± 9.03 × 10^−4^	-	-
PPE	-	-	-	LN, 7.98 × 10^−3^ ± 3.25 × 10^−3^
MFE	-	LN, 7.51 × 10^−3^ ± 6.05 × 10^−3^	LN, 1.25 × 10^−1^ ± 8.98 × 10^−2^	-
Total risk value	LN, 6.38 × 10^−1^ ± 3.49 × 10^−1^	LN, 3.76 × 10^−1^ ± 1.72 × 10^−1^	LN, 5.36 × 10^−1^ ± 3.17 × 10^−1^	LN, 1.72 × 10^−1^ ± 5.74 × 10^−1^

LN: Lognormal distribution (parameter values are mean value ± standard deviation).

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
