# Peer review of "Risk Assessment of Miners’ Unsafe Behaviors: A Case Study of Gas Explosion Accidents in Coal Mine, China"

_ijerph, 2019, doi:10.3390/ijerph16101765_

Round 1

Reviewer 1 Report

The paper describes and analyzes the safety situation in the Chinese coal mines.

The impression  you give in the Introduction is that the situation in the mines worsens. But is that really the case? Or is it a matter of perception and are a few fatalities today felt as equally terrible with a hundred ones years ago? Can you show a statistic of fatalities per year?

A general comment is that throughout the paper there seems to be a lack of understanding that behavior of workers depends on safety climate. In turn, a good safety climate is determined by management, and in fact, the attitude of the top leader (CEO) determines safety climate and with that the safety culture. So, improvement is in the hands of the top management! A first measure would be to introduce a safety management system.

Lines 67-69 What do you mean with the sentence with respect to "personal risks"?
Line 82: What do you mean with "type" of miners?
Line 93: Working types or types of work?
Line 102: Ventilation: what functions will mine people have with respect to ventilation? Maintenance? Control of functioning?
Table 2: Unsafe behaviors? Many entries in the table are parts of equipment. So, these should be formulated differently.
Line 168. Adventure should be Venture.

The background/foundation of equation (1) is unclear.

How did you derive the figures in Tables 4 and 5?

There is a problem of dependency in equations (3) and (4). More correct would be: P is the probability of unsafe behavior, then B must be the probability of an accident with severity C given an unsafe behavior.

Line 265: Didn't we know that already? What is new in the prediction?

Line 280: Why are you performing a MC routine, I cannot see the added value? What do you want to prove? That ventilation is the highest fraction of mine accident you knew already. So far, the risk assessment has been descriptive (based on accident data); the purpose of risk assessment is prediction. Purpose of your work could be to predict a risk reduction by certain measures in general (safety culture improvement) or specific (equipment or procedure improvement) given the generated knowledge of descriptive risk.

The English is not that bad but is not everywhere correct. Check by a linguistic expert is recommended.

Conclusions must be adapted to text changes as a result of a revision.

Author Response

Response to Reviewer 1 Comments

Point 1: The paper describes and analyzes the safety situation in the Chinese coal mines. The impression you give in the Introduction is that the situation in the mines worsens. But is that really the case? Or is it a matter of perception and are a few fatalities today felt as equally terrible with a hundred ones years ago? Can you show a statistic of fatalities per year?

Response 1: Thank you for this constructive advice. We seriously considered your comments and made some corrections. First, we changed the title of the article to “Risk Assessment of Miners’ Unsafe Behaviors: A Case Study of Gas Explosion Accidents in Coal Mine, China”, and the content of the article is also selected from the coal mine accident in China. Second, the global coal mine situation is getting better, and we have added some descriptions in the introduction. Third, since this paper studies the gas explosion accidents in China's coal mines, we will show a statistic of fatalities per year of gas explosion accidents from 2001 to 2018. More detailed revisions are shown as follows.

1. “Risk Assessment of Miners’ Unsafe Behaviors: A Case Study of Gas Explosion Accidents in Coal Mine, China” (Lines 3, Page 1)

2. “Not only that, Coal mine accidents around the world have been well controlled and prevented, and many countries have adopted advanced coal mine production equipment and technology to ensure the safety of miners. However, for coal mine production accidents in China, gas explosion accidents are still a prominent problem in the field of coal mine safety, which occupy an absolute proportion in major and special mine accidents [17-18].” (Lines 59-64, Page 2)

3.

Figure 2. Statistics of gas explosion accidents from 2001 to 2018

“This paper statistics data on the occurrence of coal mine accidents in China from 2001 to 2018 and presents them in the form of a histogram. It can also be seen in Figure 2 that the gas explosion accident has been declining year by year. This is because China has attached great importance to mine accidents and invested more advanced equipment and technology.” (Lines 56-59, Page 2)

Point 2: A general comment is that throughout the paper there seems to be a lack of understanding that behavior of workers depends on safety climate. In turn, a good safety climate is determined by management, and in fact, the attitude of the top leader (CEO) determines safety climate and with that the safety culture. So, improvement is in the hands of the top management! A first measure would be to introduce a safety management system.

Response 2: We strongly agree with your suggestion. In the section of "4.4. Unsafe Behavior Risk Management", we added descriptions of factors such as safety culture, safety climate and safety management system, and explained that the introduction of safety management system is the primary measure. More detailed revisions are shown as follows.

“As far as we know, for coal mining companies, safety culture and safety climate are indispensable support for the continuous realization of safety production. Due to only relying on scientific and technological means can’t achieve real safety, safety culture and scientific management methods are needed to make workers clear their responsibilities and prevent accidents. Scientific and effective safety management methods, specific safety management objectives, favorable safety production environment, safety culture and safety climate form a safety management system. Perfect safety management system is constructed by management and workers, which function is to improve the production performance of enterprises and reduce the occurrence of unsafe behavior.” (Lines 400-409, Page 16)

Point 3: Lines 67-69 What do you mean with the sentence with respect to "personal risks"?

Response 3: We are very sorry for the misunderstanding of the reviewers due to unclear expression, and we have replaced "personal risks" with "risk of each working type".

“However, many studies only consider the assessment of unsafe behaviors of coal mine system, but do not involve risk of each working type [28-29].” (Lines 82-83, Page 3)

Point 4: Line 82: What do you mean with "type" of miners?

Response 4: According to your suggestion, we changed "type" to "working types". Because in the coal mine, many workers are engaged in different jobs, while some workers are engaged in the same type of job, and each kind of work is called "working types".

“Coal mine production involves many different working types of miners, but at present there are no specific standards and regulations for the division of working types in the coal industry.” (Lines 96, Page 3)

Point 5: Line 93: Working types or types of work?

Response 5: We think it should be expressed as "working types". The occurrence of gas explosion accidents is related to many posts, meanwhile, each working types includes multiple posts.

“Table 1. Statistics of working types associated with gas explosion accidents.” (Lines 107, Page 3)

Point 6: Line 102: Ventilation: what functions will mine people have with respect to ventilation? Maintenance? Control of functioning?

Response 6: We added the work functions of the ventilation working type to the section of “2.2.1. Unsafe Behaviors of Key Working Types”.

The revised contents are as follows.

“The main functions of the ventilation working type are to monitor the operation and maintenance of the ventilator in time and pay attention to the air volume in the roadway. Attention should be paid to the formulation of mine ventilation measures to avoid the occurrence of unsafe behavior.” (Lines 124-127, Page 4)

Point 7: Table 2: Unsafe behaviors? Many entries in the table are parts of equipment. So, these should be formulated differently.

Response 7: Based on your comment, we changed the title of Table 2 to "Table 2. Influencing factors of unsafe behaviors in key working types". (Lines 176, Page 5)

Point 8: Line 168. Adventure should be Venture.

Response 8: We have changed "Adventure" to "Venture". Meanwhile, the abbreviation "ADP" for the sixth type of unsafe behavior was replaced by "VDP". The content related to the abbreviation of the sixth unsafe behavior in the article was all changed to "VDP".

“(6)       Venture into dangerous places (VDP);” (Lines 185, Page 6)

Point 9: The background/foundation of equation (1) is unclear.

Response 9: Because of our mistakes, the background of formula (1) is not clearly marked. Formula (1) is calculated and integrated according to reference [45].

“Similarly, based on the research foundation of Dan et al [45], the model of the relationship between the influencing factors is obtained,…” (Lines 222, Page 8)

Point 10: How did you derive the figures in Tables 4 and 5?

Response 10: For the figures in Table 4 and Table 5, we mainly refer to the information in document [24] [47] and combine with some questionnaires about the actual working conditions of Chinese coal miners in different jobs. (Lines 232-233, Page 9)

Point 11: There is a problem of dependency in equations (3) and (4). More correct would be: P is the probability of unsafe behavior, then B must be the probability of an accident with severity C given an unsafe behavior.

Response 11: We strongly agree with your comments and have made the following amendments.

“Where R is the risk of unsafe behavior, P is the revised probability of unsafe behavior, I is accident hazard index, C is the severity of accident losses, B is the probability of an accident with serious consequence C caused by unsafe behavior.” (Lines 272-274, Page 10)

Point 12: Line 265: Didn't we know that already? What is new in the prediction?

Response 12: For the questions you have raised, we have removed the obvious results and added possible reasons that would lead to the results in Table 8. Specific revisions are shown as follow.

“From the results in Table 8, it can be seen that ventilation is the most likely type of work to cause gas explosion accidents, both from the point of view of the occurrence of unsafe behavior and from the perspective of the revised occurrence probability. The possible reason for the results in the table is that ventilation is the most important factor to prevent gas explosion in coal mines. Ventilation problems can occur due to ventilating equipment, unsafe behavior of workers and so on. The number of unsafe behaviors in gas prevention and fire extinguishing ranks second. Because the basic element of the gas explosion accident is the ignition source, and the workers of gas prevention and fire extinguishing working type have imperfect preventive measures, which will very likely to lead to gas explosion accident. The probabilities of unsafe behavior in blasting and electrician are relatively small. The reason for this difference in results may be that the blasting equipment and electrical equipment in modern coal mines are relatively advanced, which can effectively guarantee the safety of coal mine production and miners.” (Lines 286-295, Page 11)

Point 13: Line 280: Why are you performing a MC routine, I cannot see the added value? What do you want to prove? That ventilation is the highest fraction of mine accident you knew already. So far, the risk assessment has been descriptive (based on accident data); the purpose of risk assessment is prediction. Purpose of your work could be to predict a risk reduction by certain measures in general (safety culture improvement) or specific (equipment or procedure improvement) given the generated knowledge of descriptive risk.

Response 13: In order to evaluate the health risks caused by unsafe behaviors more scientifically and accurately, we developed a probabilistic risk assessment model based on Monte-Carlo simulation. The essence of this probabilistic risk assessment model is to repeat sampling from several input variable probability distributions to establish the distribution of output variables.

Point 14: The English is not that bad but is not everywhere correct. Check by a linguistic expert is recommended.

Response 14: We have repeatedly confirmed the expression and grammar of the article.

Point 15: Conclusions must be adapted to text changes as a result of a revision.

Response 15: The conclusions have been adjusted based on the revised text.

“More importantly, scientific and effective safety management methods, specific safety management objectives, favorable safety production environment, safety culture and safety climate are all important factors in the formation of a sound safety management system. Constructing a complete safety management system is an important measure to effectively improve the performance of coal enterprises and reduce the occurrence of miners’ unsafe behaviors.” (Lines 445-451, Page 17)

Reviewer 2 Report

The article is well-written and organized.
I recommend a few minor changes before accepting the journal, which must be included in the article.

The hard coal is not extracting only in China. Gas explosions are a serious threat in all hard coal mines, in China, in Poland, in Russia. The title of the article does not apply only to China. The importance of the problem should therefore be emphasized.

1.     Line 13/14 „This paper  selects 200 gas explosion (…)” -  Paper select? In my opinion, the authors select accidents, which research in this paper.

2.     Coal is not only in China an important energy resource. Also, e.g. in Poland. This should also be emphasized.

3.     After this sentence„Among the many types of accidents, gas explosion accidents are considered to be the most serious type of accident in  coal mine accidents.”, please add references:

 -       Brodny, J.; Tutak, M. Analysis of methane hazard conditions in mine headings. Tehn. Vjesn. 2018, 25, 271–276, doi:10.17559/TV-20160322194812.

-       Krause, E.; SmoliĹ„ski, A. Analysis and Assessment of Parameters Shaping Methane Hazard in Longwall Areas. J. Sustain. Min. 2013, 12, 13–19.

-       Tutak, M.; Brodny, J. Analysis of the Impact of Auxiliary Ventilation Equipment on the Distribution and Concentration of Methane in the Tailgate. Energies 2018, 11, 3076.

-       Tutak, M.; Brodny, J. Analysis of Influence of Goaf Sealing from Tailgate on the Methane Concentration at the Outlet from the Longwall. IOP Conf. Ser. Earth Environ. Sci. 2017, 95, 042025, doi:10.1088/1755-1315/95/4/042025.

-       Trenczek, S. Methane ignitions and explosions in the context of the initials related to technical and natural hazards. Przegl. GĂłrniczy 2015, 72, 87–92.

4.     Aftere sentence “Based on the research of Dan et al,” you need added reference.

Kind regards, 

Reviewer

Author Response

Response to Reviewer 2 Comments

The article is well-written and organized.

I recommend a few minor changes before accepting the journal, which must be included in the article.

Response: The authors would like to thank the editor and reviewers for their efforts and valuable comments. At the same time, we are very grateful to the reviewers for their affirmation.

Point 1: The hard coal is not extracting only in China. Gas explosions are a serious threat in all hard coal mines, in China, in Poland, in Russia. The title of the article does not apply only to China. The importance of the problem should therefore be emphasized.

Response 1: We have accepted your suggestion and made an appropriate change to the title of the article, while adding a description of the international coal mine situation in the section of introduction. Specific revisions are shown as follows.

1. The title of the article was changed to “Risk Assessment of Miners’ Unsafe Behaviors: A Case Study of Gas Explosion Accidents in Coal Mine, China”. (Lines 3, Pages 1)

2. “Coal mines have always playing a vital role in promoting the development of various countries. It is an important energy source for countries in the world.” (Lines 29-30, Pages 1)

3. In addition, we have added a brief description of the global coal mine situation, add a new histogram was added to express the annual number of coal mine gas explosion accidents in China.

“This paper statistics data on the occurrence of coal mine accidents in China from 2001 to 2018 and presents them in the form of a histogram. It can also be seen in Figure 2 that the gas explosion accident has been declining year by year. This is because China has attached great importance to mine accidents and invested more advanced equipment and technology. Not only that, Coal mine accidents around the world have been well controlled and prevented, and many countries have adopted advanced coal mine production equipment and technology to ensure the safety of miners. However, for coal mine production accidents in China, gas explosion accidents are still a prominent problem in the field of coal mine safety, which occupy an absolute proportion in major and special mine accidents [17-18].” (Lines 55-63, Pages 2)

Point 2: Line 13/14, This paper selects 200 gas explosion (…)” - Paper select? In my opinion, the authors select accidents, which research in this paper.

Response 2: Thank you for this suggestion, which can significant improve the clarity of our expressions. We have changed the "Paper select" in the text to "the authors select". (Lines 13-14, Pages 1)

Point 3: Coal is not only in China an important energy resource. Also, e.g. in Poland. This should also be emphasized.

Response 3: As you said, coal mines are not only an important energy resource in China, but also an irreplaceable role in other countries. We add this part of the description to the section of introduction, as amended below.

“Not only for Chinese coal mines, but also for many other countries, gas explosion is also a serious threat [11-15]. For example, mining is a very important industry in Poland. One of the most common hazards in a coal mine accident is the methane hazard, so the fatalities rate of gas explosion accidents is very high. When the situation is serious, it may account for half of the number of coal mine accidents in the same period. Similarly, gas explosion accidents in countries such as Russia and Ukraine are also serious.” (Lines 43-48, Pages 1-2)

Point 4: After this sentence, Among the many types of accidents, gas explosion accidents are considered to be the most serious type of accident in  coal mine accidents.”, please add references.

Response 4: The articles you provide are very helpful for our research on gas explosion accidents, and we have already cited these references in the article.

Point 5: After sentence “Based on the research of Dan et al,” you need added reference.

Response 5: Due to our negligence, the reference was not indicated. We have already indicated the reference [45] after "Based on the research of Dan et al," and the revised content is as follows.

“Based on the research of Dan et al [45], this paper divides the influencing factors of coal mine workers' unsafe behaviors into organizational factors, human-machine factors and personal factors.” (Lines 214, Pages 8)

Round 2

Reviewer 1 Report

Most previous comments have conceptually been responded to satisfactorily, although the English in the new parts needs improvement. Word are missing, e.g., line 29 'been' in have been playing; line 56 'shows' in This paper shows statistics.
Looking at what you actually produced I would not call it a risk assessment as these are almost always predictive, but you performed a risk analysis based on existing data, so the study is descriptive. Therefore if you replace in the title of the paper the word 'assessment' by 'analysis', a reader might not become disappointed and the appreciation for the paper will increase.

Author Response

Response to Reviewer 1 Comments

Point: Most previous comments have conceptually been responded to satisfactorily, although the English in the new parts needs improvement.

Response: The authors would like to thank the editor and reviewers for their efforts and valuable comments. At the same time, we will carefully check the English expression again.

Point 1: Word are missing, e.g., line 29 'been' in have been playing; line 56 'shows' in This paper shows statistics.

Response 1: Some words were lost due to our negligence. We have made some modifications, and also revised some other sentences and tenses in this article. More detailed revisions are shown as follows. (The second revisions of the article were marked in blue.)

1. “Coal mines have been always playing a vital role in promoting the development of various countries.”

2. “This paper shows statistics data on the occurrence of coal mine accidents in China from 2001 to 2018 and presents them in the form of a histogram.”

3. “The reasons for this situation were that more attention has been paid to mine accidents, and more advanced equipment and technology have been invested in China.”

4. “The above studies show that it must be considered from the perspective of human unsafe behaviors in order to prevent gas explosion accidents”.

5. “The authors studied the unsafe behaviors of workers related to gas explosion from the perspective of working types, and listed the above six working types.”

6. “the influencing factors of coal mine workers' unsafe behaviors were divided into organizational factors,”

7. “coal mine enterprises should improve working environment and change workload according to the working characteristics of different working types.”

……

Point 2: Looking at what you actually produced I would not call it a risk assessment as these are almost always predictive, but you performed a risk analysis based on existing data, so the study is descriptive. Therefore if you replace in the title of the paper the word 'assessment' by 'analysis', a reader might not become disappointed and the appreciation for the paper will increase.

Response 2: We greatly admire your earnest attitude towards scientific research and thank you for your valuable comments. But we consider that the expression "risk assessment" is also appropriate. In this article, we have established a risk assessment model and used probabilistic risk assessment methods to quantify the risk of unsafe behavior that may lead to gas explosion accidents. The above content is actually a more accurate risk prediction process. The probabilistic risk assessment method can reduce the uncertainty of risk. Meanwhile, the sensitivity analysis of the parameters affecting the risk results was carried out by using Crystal Ball software, which can put forward the improvement measures more pertinently. All of the above processes belong to risk assessment. The scope of "risk assessment" is relatively small compared with "risk analysis", which can summarize our work more clearly and accurately.
